# Low-Cost Detection of Methane Gas in Rice Cultivation by Gas Chromatography-Flame Ionization Detector Based on Manual Injection and Split Pattern

**DOI:** 10.3390/molecules27133968

**Published:** 2022-06-21

**Authors:** Chaofeng Li, Qingge Ji, Xianshu Fu, Xiaoping Yu, Zihong Ye, Mingzhou Zhang, Chuanxin Sun, Yulou Qiu

**Affiliations:** Zhejiang Provincial Key Laboratory of Biometrology and Inspection & Quarantine, College of Life Science, China Jiliang University, Hangzhou 310018, China; lcf609162059@163.com (C.L.); qinggeji1997@163.com (Q.J.); yxp@cjlu.edu.cn (X.Y.); zhye@cjlu.edu.cn (Z.Y.); zmzcjlu@cjlu.edu.cn (M.Z.); chuanxin.sun@slu.se (C.S.); yulou@cjlu.edu.cn (Y.Q.)

**Keywords:** rice cultivation, methane, gas chromatography-flame ionization detector (GC-FID), manual injection, split ratio (SR)

## Abstract

Rice cultivation is one of the most significant human-created sources of methane gas. How to accurately measure the methane concentration produced by rice cultivation has become a major problem. The price of the automatic gas sampler used as a national standard for methane detection (HJ 38-2017) is higher than that of gas chromatography, which greatly increases the difficulty of methane detection in the laboratory. This study established a novel methane detection method based on manual injection and split pattern by changing the parameters of the national standard method without adding any additional automatic gas samplers. The standard curve and correlation coefficient obtained from the parallel determination of methane standard gas were y = 2.4192x + 0.1294 and 0.9998, respectively. Relative standard deviation (RSD, <2.82%), recycle rate (99.67–102.02%), limit of detection (LOD, 0.0567 ppm) and limit of quantification (LOQ, 0.189 ppm) of this manual injection method are satisfying, demonstrating that a gas chromatography-flame ionization detector (GC-FID), based on manual injection at a split ratio (SR) of 5:1, could be an effective and accurate method for methane detection. Methane gases produced by three kinds of low-methane rice treated with oxantel pamoate acid, fumaric acid and alcohol, were also collected and detected using the proposed manual injection approach Good peak shapes were obtained, indicating that this approach could also be used for quantification of methane concentration.

## 1. Introduction

Nowadays, the problem of global warming caused by the increasing concentration of Greenhouse Gases (GHGs) in the atmosphere is becoming more and more serious. GHGs not only have a negative impact on crop production but also pose a threat to world food production and supply [1]. According to the estimates of the Intergovernmental Panel on Climate Change (IPCC, Geneva, Switzerland), surface temperature has increased by 0.6 ± 0.2 °C over a century, but if GHG emissions (mainly carbon dioxide (CO_2_), methane (CH_4_) and nitrous oxide (N_2_O)) continue to increase, the global average temperature will rise by 0.2 °C every 10 years [2]. At the same time, the IPCC points out that agricultural GHG emissions are very similar to that of transport GHGs, accounting for 13.5% of the global total [3]. As the second most important greenhouse gas after CO_2_ [4,5], methane accounts for about 20% of the total global warming forcing [6]. According to relevant materials, about 50–65% of CH_4_ in the atmosphere is anthropogenic or human-generated [7,8], most of which comes from rice cultivation fields. In the absence of sufficient oxygen, microbes living in rice roots produce methane, which is absorbed, along with water, by the rice roots and transported to the stems and leaves before it eventually escapes into the atmosphere. Authoritative statistics show that the annual methane emissions from rice fields are about 3.1 × 10^10^–11.2 × 10^1^^0^ kg, accounting for 7–17% of atmospheric methane [9,10,11]. Data show that on a 100-year time scale, the global warming potential (GWP) of unit mass of CH_4_ is 28 times higher than that of CO_2_, contributing about 15% to the greenhouse effect [2,12,13,14]. Since rice is one of the basic food crops of the world [15,16] and the most important food crop in China, the world’s rice planting area is as high as 17.8% [17,18]. In addition, as the world’s population continues to grow, the demand for rice is also expected to increase by 35% by 2030 [19,20,21,22]. Therefore, with the increase of rice cultivation, how to effectively reduce the GHG emissions in the process of rice cultivation has gradually become a hot concern for scholars all over the world [23]. For example, Qi Le et al. [24] concluded that the application of biochar would reduce the abundance of methanogenic archaea in flooded paddy soil, thus reducing the emission of CH_4_. In that research, the GC was connected to an automatic air intake device, allowing direct measurement of methane gas collected by the automatic intake device. Jieyun Liu et al. [25] also analyzed methane using an automatic gas inlet connected to the GC, and found that, compared with the control group, the addition of biochar to granite weathered red soil and Quaternary red clay reduced methane emissions by 19.8–28.2% and 31.7–37.1%, respectively. At the same time, increasing soil pH value and reducing soil NH_4_^+^–N are beneficial to decrease CH_4_ emission. Pengfu Hou et al. [26] collected methane using an automatic sampling device and directly detected methane concentration through a GC. The study found that the addition of sawdust-derived hydrogenated coke could not increase methane emissions due to the limitation of the matrix, and its cumulative emission was similar to the control, ranging from 11.1 g m^−2^ to 12.8 g m^−2^. Yanfang Feng et al. [27] determined that mixed application of wood vinegar and biochar could significantly diminish methane emissions through the autosampler sending the collected methane gas to the GC for detection. Similar to this, Tiehu He et al. [28] used the autosampler to input the collected methane gas into the GC for detection and confirmed that biochar combined with urease and inhibitors can cut down methane emissions from paddy fields. The above researchers used GC-FID and automatic injection to detect the methane gas collected during the experiment, which is the most common method. In order to accurately determine the effects of different treatments on methane emissions, it is extremely important to conveniently and accurately measure the methane content collected during rice growth.

At present, China has a set of standards for the detection of methane in ambient air, and the National Environmental Protection Standard of the People’s Republic of China HJ 38-2017 is one of them [29]. This standard stipulates the quantitative analysis of methane concentration using gas chromatography-flame ionization detection (GC-FID) in conjunction with an automatic gas sampling device (about 400,000 RMB), which is more expensive than the gas chromatography device itself (about 300,000 to 400,000 RMB). The purpose of this study is to propose a manual injection method that could accurately detect methane gas concentration by changing the parameters on the basis of National Environmental Protection Standard. This reduces the cost requirements of ordinary laboratories to detect the concentration of methane gas and allows more laboratories to better explore ways to produce low-methane rice, thereby lowering anthropogenic GHG emissions.

## 2. Results and Discussion

### 2.1. Qualitative Analysis of Splitless Mode

The capillary columns (HP-MOLESIEVE) used in this study are special columns for methane detection. The other component in standard gas is nitrogen, which is non-detectable by FID detector, therefore, the chromatogram should show only a single methane response peak.

The column chamber temperatures were 60 (Figure 1a), 70 (1b), 80 (1c), 90 (1d) and 100 °C (1e), respectively, to clarify the influence of different column chamber temperatures on chromatographic peaks. The chromatograms are shown in Figure 1. As can be seen from Figure 1, the peak time decreased with the increasing of temperature of the column chamber in the manual and splitless injection mode, but the ghost peaks in the chromatogram were still not ameliorated. Changing and altering the detector temperature (Figure 2a) and the gasification chamber temperature (Figure 2b) also had no effect on improving the chromatographic peak shape. Figure 3 is the chromatogram obtained after changing the make-up gas parameters. In Figure 3a–f, the make-up gas decreased from 24 mL min^−^^1^ to 19 mL min^−^^1^ at intervals of 1 mL min^−^^1^. As shown in Figure 3, when the make-up gas flowrate was altered, the peak time diminished with the decrease of the make-up gas flowrate, but the chromatogram was still not theoretically unimodal. It can be seen that in the case of manual injection, the above four parameters were not the factors leading to the chromatogram ghost peaks There may be two reasons for the appearance of ghost peaks. The most likely reason is that manual injection could not be as fast, uniform and accurate as automatic injection, making it impossible for all the methane gas in the manual injection method to enter the capillary column at the same time. There was a batch entry situation of the methane gas, which eventually led to the phenomenon of ghost peak. Another reason may be that the amount of gas injection exceeded the capacity of the capillary column, which also caused the methane gas to enter the detector in batches, resulting in ghost peaks. The common reason for both was that all of the gas sample couldn’t pass through the capillary column into the detector at the same time.

### 2.2. Qualitative Analysis of Split Mode

So as to explore whether the split pattern could modify the ghost peak situation, we changed the manual injection mode from splitless to split pattern, and the temperatures of column, gasification chamber and detector were 80, 100 and 200 °C, respectively. At this time, the make-up gas flow rate was controlled at 22 mL min^−1^. Adopting Formula (1), when SR was 1:1, 2:1, 3:1, 4:1 and 5:1, respectively, the concentration of 16 ppm standard methane gas was detected by manual injection and each ratio repeated 5 times. As in Figure 4, in split mode, after entering the liner, a part of the gas sample was discharged with the carrier gas through the split valve, and the remainder entered the detector along the chromatographic column. Through the split vent, the carrier gas with the sample was split, which allowed the right amount of sample to enter the capillary column for separation and analysis.

In order to improve the chromatographic peak shape, the amount of methane gas entering the capillary column was reduced by changing the manual splitless mode to a manual split one, so that all methane gas could enter the detector at one time. Figure 5 showed the chromatograms of 16 ppm standard methane gas at five different SRs. It can be seen from Figure 5 that as SR increased from 1:1 to 5:1, the retention time did not change significantly, remaining at a relatively fixed value, but the half-peak width gradually decreased. Under the conditions of 80 °C column temperature, 100 °C gasification chamber temperature, 200 °C detector temperature and 22 mL min^−^^1^ make-up gas, the retention time was 1.330–1.327 min, and the peak time did not change significantly at about 1.3 min.

### 2.3. Quantitative Analysis of Split Mode

The relationship between the methane equivalent content in standard gas and the peak area is shown in Figure 6, where linear fitting of the standard curve was y = 2.4192x + 0.1294, and the correlation coefficient (R^2^) was 0.9998 (>0.995), indicating that the manual injection approach was feasible.

Derived from the workstation OpenLAB CDS, when the split ratio was 1:1, 2:1, 3:1, 4:1, 5:1 and 6:1, the corresponding symmetry factors of the peak shape were 0.861, 0.923, 0.929, 0.942, 0.985 and 1.051, respectively. It is considered that the larger the SR is, the less samples will enter the detector and the lower the peak height will be, which put forward higher requirements for instrument sensitivity. According to the symmetry factor values under the above different SRs, the symmetry factor with an SR of 5:1 was the best value; therefore an SR of 5:1 was selected.

### 2.4. Method Calibration and Validation

#### 2.4.1. Accuracy and Recycle Rate

In order to judge the accuracy and precision of the methane detection method, the method was used to detect methane standard gas concentrations of 5 ppm, 10 ppm, 16 ppm, 20 ppm and 25 ppm for 5 parallel determinations. The relative standard deviation (RSD) was calculated to verify the precision of the proposed method. At the same time, the recycle rate, that is, the ratio of measured content and standard content, was calculated to evaluate the accuracy of the method. Results of RSD and recycle rate were shown in Table 1. For the detection of 5, 10, 16, 20 and 25 ppm methane gas, the RSD was 2.19, 1.60, 2.71, 2.82 and 1.01%, respectively, and the recovery range was 99.67–102.02% [30].

To ensure the precision and accuracy of the method, more work is required during sampling. First, the R^2^ of the standard curve should be greater than or equal to 0.995. Second, at least 10% of each batch of samples should be replicated. Simultaneously, the relative deviation of the measurement results should not be greater than 15% [29].

#### 2.4.2. Method Detection Limit

In order to evaluate the proposed methane test method, linearity, limit of detection (LOD, signal-to-noise ratio of 3:1) and limit of quantification (LOQ, signal-to-noise ratio of 10:1) were studied. Standard methane gases with concentrations of 1 ppm, 2 ppm and 3 ppm were measured in parallel 5 times, and the standard deviations of each concentration point were recorded. Taking the standard deviation as the ordinate and the gas concentration as the abscissa, the results are shown in Figure 7. The linear regression equation, y = –0.0049x + 0.0189, was obtained. The standard deviation of zero concentration was obtained by extrapolation and identified as S_0_, which is the signal-to-noise ratio. LOD and LOQ of method were defined as 3S_0_ and 10S_0_ [31,32], which were 0.0567 and 0.189, respectively. The results are shown in Table 2.

### 2.5. Actual Gas Samples Detection

Figure 8 showed the chromatogram of the measured methane gas concentration produced by low-methane rice. The concentrations of methane emissions of low-methane rice treated with F, O and AC treatment were 33.04208 (Figure 8a), 12.61878 (Figure 8b) and 172.73617 ppm (Figure 8c), respectively. It can be seen that the method had good detection effects on real samples under three different treatments.

## 3. Materials and Methods

### 3.1. Materials

16 ppm and 25 ppm of methane standard gases were bought from Hangzhou New Century Mixed Gas Co., Ltd. (Hangzhou, China). All glass syringes were obtained from the Shanghai Chemical Experimental Equipment Co., Ltd. (Shanghai, China). Tedlar polyvinyl fluoride (PVF) sampling bags with capacity of 1 L was supplied by Delin Gas Packing Co., Ltd. (Dalian, China). The manual injection needle (P/N5190-1531) was supplied by the Agilent Singapore factory. The actual methane samples were collected from a low-methane paddy field. Low-methane rice was the product of rice treated with oxantel pamoate treatment (O treatment), fumaric acid treatment (F treatment) and alcohol treatment (AC treatment). Low-methane rice was planted in experimental fields, and methane gases were collected at different growth stages and sunshine duration as required. 30 methane gas samples were gathered from low-methane rice for each treatment method and manually injected for detection.

### 3.2. Experiment Methods for Apparatus and GC Analysis

The GC (7890B, Agilent, Singapore) used in this study was equipped with flame ionization detector (FID) and capillary column (HP-MOLESIEVE, 0.530 mm × 30 m × 25.0 μm, Agilent, Singapore). The workstations for software manipulation and data processing were OpenLAB CDS developed by Agilent. According to the referenced standard (HJ 38-2017) [30], the temperature of the column chamber, gasification chamber and detector was 80 °C, 100 °C and 200 °C, respectively. Nitrogen (99.999%, Min Xing Gong Mao Co., Ltd., Hangzhou, China) was used as the carrier gas, and the total flow rate was set at 30 mL min^−^^1^. The carrier gas was then divided into two channels, in which the flow of the make-up gas was controlled at 22 mL min^−^^1^, and the other flow of the capillary column was controlled at 8 mL min^−^^1^. Air flow (99.999%, Min Xing Gong Mao Co., Ltd., Hangzhou, China) and hydrogen flow (99.999%, Jin Gong Special Gas Co., Ltd., Hangzhou, China) were set at 300 mL min^−^^1^ and 30 mL min^−^^1^, respectively. The automatic injection pattern mentioned in the standard was replaced by manual injection based on the existing conditions in the laboratory.

### 3.3. Preparation of Standard Gas

The explosion limit of methane is 4.4–17%, or 44,000 to 170,000 parts per million (ppm), which is much higher than standard methane gas with a concentration of 25 ppm. Standard methane gas concentrations of 16 ppm and 25 ppm were diluted by the syringe gas distribution method [33]. First, two syringes were used to extract 80 mL of standard methane gas and 20 mL of high-purity nitrogen gas, and then the gases from the two syringes were injected into 100 mL blank syringes with rubber tubes successively. After fully mixing, a gas with 20 ppm concentration of standard gas was obtained. According to this method, eight gradient standard methane gases with concentrations of 1, 2, 3, 5, 10, 16, 20 and 25 ppm were finally acquired.

### 3.4. Exploring the Conditions in the Splitless Mode

Appropriate temperature conditions and make-up gas flow rate under manual injection in splitless mode were explored. Temperature parameters such as column chamber temperature, gasification chamber temperature and detector temperature, would significantly affect the separation quality of GC. The column chamber temperature could affect the pressure at the front of the column and the flow rate of the carrier gas. Taking this study as an example, the column chamber temperature affected both the retention time of methane gas and the peak type of the chromatographic peak. Under normal circumstances, the higher the chamber temperature is, the shorter the retention time will be and the narrower the half peak width is, the higher the peak height will be, and vice versa. In general, the retention time of each component would change by 5% when the column chamber temperature altered by 1 °C. The temperature of the gasification chamber also has an impact on factors such as retention time, half-peak width, peak height and so on. When the gasification chamber temperature changed from low to high, the retention time and half-peak width decreased, and the peak height increased first and then stabilized. As well as the temperature of the column chamber and gasification chamber, the temperature of the detector also had a significant influence on the results of GC. If the temperature was too high, the response value and baseline noise of the components would increase significantly, reducing the sensitivity of the instrument.

The make-up gas was usually a branch of carrier gas, which entered the detector directly from the rear end of the chromatographic column in GC. Since there was a dead volume at the connection between chromatographic column and detector, this dead volume would have a serious effect on the peak shape and the efficiency of capillary column. The adverse influence of dead volume on the peak shape could be eliminated by rapidly blowing the sample components into the detector. Therefore, it was strongly necessary to set the make-up gas of the capillary column. Traditionally, the sum flow rate of carrier gas and make-up gas of the capillary column in FID detector is about 30 mL min^−^^1^ [34,35,36]. Small make-up gas flow could improve the sensitivity, but it was prone to peak tail, affecting the peak shape, and vice versa.

In short, sample separation was not only affected by the temperature of the column chamber, gasification chamber and detector, but was also closely related to the value size of the make-up gas. Therefore, the detection and analysis of the standard methane gas concentration was carried out according to the above four key parameters and in combination with HJ 38-2017. The parameters in HJ 38-2017 are: column chamber temperature = 80 °C, gasification chamber temperature = 100 °C, detector temperature = 200 °C and make-up gas flow rate = 22 mL min^−^^1^. Only one parameter was changed during each test, and the remaining three parameters were fixed. Every parameter was repeated 5 times by manual injection to investigate the influence of each parameter on the detection results. For example, when the gasification chamber temperature, the detector temperature and the make-up gas flow were 100 °C, 200 °C and 22 mL min^−^^1^, respectively, the column chamber temperature was increased from 60 °C to 100 °C at 10 °C intervals (including 80 °C). The variables involved in the four parameters were as follows: column chamber temperature (60, 70, 80, 90 and 100 °C); gasification chamber temperature (80, 90, 100, 110 and 120 °C); detector temperature (180, 190, 200, 210 and 220 °C); make-up flow rate (24, 23, 22, 21, 20 and 19 mL min^−^^1^).

### 3.5. Exploring the Conditions in the Split Mode

When using gas chromatography to detect samples, there are usually two injection modes: split and splitless. Figure 4 is the schematic diagram of split/splitless injection of the GC inlet. As can be seen from Figure 4, when the split valve is off and the sample gas enters the liner, all the samples get in the capillary column and finally flow into the detector, which is the splitless mode. If the split valve is on, the sample gas entering the liner will be divided into two parts, one part into the capillary column and the other part into the exhaust port through the split vent. The split ratio can be adjusted by the column head pressure control, which changes the volume amount of sample entering the detector.

The split ratio (SR) is calculated using Formula (1):(1)$$\mathrm{SR}=\frac{\;SVF}{CF}$$
where SR, *SVF* and *CF* are split ratio (SR), split vent flow (red arrow in Figure 4) and column flow (blue arrow in Figure 4), respectively. SR can be set by the operating software of the gas chromatography, and the flow is allocated according to the set value of SR. During manual injection and split model, the sample entering the gasification chamber will be mixed with nitrogen gas and then divided into *SVF* and *CF* parts. The amount of sample getting into the capillary column depends on the preset split ratio (SR). The larger the split ratio is, the less the amount of sample flowing into the capillary column will be. The higher SR is, the more accurate the accuracy of quantitative results will be, and the sharper the corresponding peak shape will be. However, when SR is too high, the detection sensitivity will be reduced due to too few samples entering the gasification chamber. High SR is not suitable for trace analysis. In the opposite case, when the sample concentration is too low to have a high sensitivity in the split pattern, the splitless mode will be considered to improve the sensitivity.

The choice of split ratio depends on the peak shape, which can be judged from the symmetry factor. When the symmetry factor is in the range of 0.95–1.05, it can be called a symmetric peak. The factor less than 0.95 is the frontier peak, while the factor greater than 1.05 is the trailing peak. The calculation formula for symmetry factor is shown in Formula 2 below. The meanings of *A* and *B* involved in Formula (2) are shown in Figure 9.
(2)$$S=\frac{A}{B}$$
where S is the symmetry factor and *A* and *B* are the distances of line segments. As shown in Figure 9, the horizontal and vertical dashed lines are the 10% peak height line and the peak vertex vertical line, respectively, which intersect at point c. The horizontal dotted line crosses the blue chromatographic peak at point a on the left and point b on the right. The distance from point a to point c is *A*, and similarly, the distance from point c to point b is *B*. The value of *A* divided by *B* is the magnitude of the peak symmetry factor.

### 3.6. Calibration and Validation Methods

The optimal SR was determined according to the symmetry factor, and the standard curve under optimal conditions was measured. Gas chromatography was used to detect five concentrations of standard methane gases (5, 10, 16, 20 and 25 ppm) three times, and the corresponding peak areas were recorded. Taking methane concentration as the X-axis and the average of the three measured peak areas as the Y-axis, a standard curve of methane concentration under this method was obtained. The precision of the method was evaluated by the relative standard deviation (RSD, %) of 5 standard methane gas concentrations measured in practice. The accuracy of the approach was calculated using the recovery test. Triplicate measurements were conducted for each sample.

## 4. Discussion

Methane gas produced during rice growth is an important factor affecting global warming, which has made low-methane rice a research hotspot in recent years. But methane detection often requires GC combined with expensive automatic gas samplers. The main purpose of this paper is to develop a low-cost and simple methane detection method. In the absence of an automatic gas sampler, methane was measured by GC with manual injection on the basis of National Environmental Protection Standard (HJ 38-2017). According to the experiment, under the condition of manual injection, if the split pattern is not adopted, no matter the changes to one or more parameters of the column chamber, gasification chamber, detector temperature and capillary column flow, a normal peak pattern cannot be obtained by GC. In the split case, we obtained that a split ratio of 5:1 was the optimal condition according to the symmetry factor. R square of standard curve (R^2^ = 0.9998) revealed that the linear relationship was good, indicating the feasibility of this method. According to the literature, the methane in the air is 1.9857 PPM [37], and the LOD and LOQ obtained by the test of 1,2,3 PPM methane gas show that this method can suit the detection of methane produced by low-methane rice. Whether this method can be used to detect high concentrations of methane in the environment, such as marsh ponds, needs to be proved experimentally.

## 5. Conclusions

The aim of this study was to establish a novel manual injection method with gas chromatography for the detection of methane produced by low-methane rice. The results showed that the concentration of methane could be accurately detected only in the split mode under manual injection mode. The optimal conditions for detection were an SR of 5:1, a column chamber temperature of 80 °C, a gasification chamber temperature of 100 °C, a detector temperature of 200 °C, and a make-up gas flow rate of 22 mL min^−^^1^. Using this method, the methane concentration obtained had a good linear relationship with its chromatographic peak area and the linear fitting function was y = 2.4192x + 0.1294. (R^2^ = 0.9998). Relative standard deviation (RSD, <2.82%), recycle rate (99.67–102.02%), limit of detection (LOD, 0.0567 ppm) and limit of quantification (LOQ, 0.189 ppm) of this manual injection method are satisfying, demonstrating that a gas chromatography-flame ionization detector (GC-FID) based on manual injection at a split ratio (SR) of 5:1 could be an effective and accurate method of methane detection.

In conclusion, in the absence of an automatic sampler, the accurate measurement of methane concentration in low-methane rice can be achieved by using the split mode based on manual injection mode. This method could reduce the hardware investment in instruments for testing methane concentration in laboratories that wanted to study low-methane rice, and enables a more concise, convenient and low-cost method of accurately measuring the methane gas produced by rice, contributing to improving global warming.

## Figures and Tables

**Figure 1 molecules-27-03968-f001:**
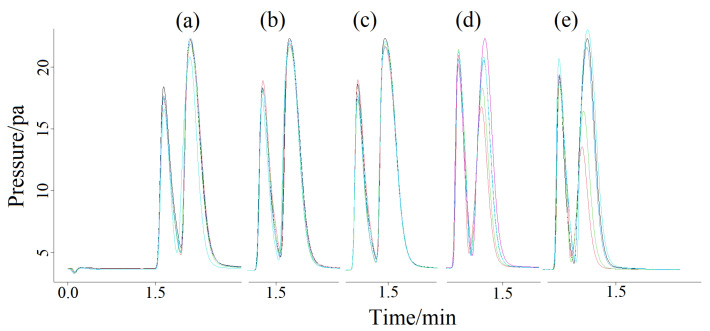
Ghost peaks of methane at different column chamber temperatures (**a**–**e**) from 60 to 100 °C (10 °C/interval, 5 repetitions per sample) based on manual splitless mode. Peak at column chamber temperature of 60 (**a**), 70 (**b**), 80 (**c**), 90 (**d**) and 100 °C (**e**), respectively.

**Figure 2 molecules-27-03968-f002:**
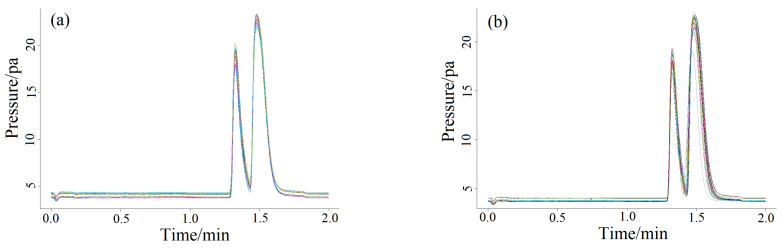
Ghost peaks of methane at different FID-detector (**a**) and gasification chamber (**b**) temperature (10 °C/interval, 5 repetitions per sample) based on manual splitless mode.

**Figure 3 molecules-27-03968-f003:**
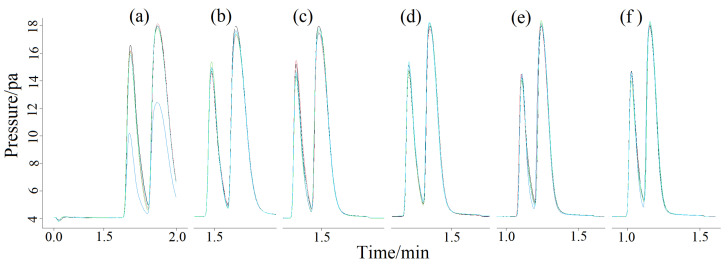
Ghost peaks of methane were measured at different flow rates of capillary column (**a**–**f**) (5 repetitions per sample, 30 mL min^−^^1^ of total carrier gas flow) in manual splitless mode. The make-up gas flow rates in the capillary column were 24 (**a**), 23 (**b**), 22 (**c**), 21 (**d**), 20 (**e**) and 19 mL min^−^^1^ (**f**), respectively, at a total flow rate of 30 mL min^−^^1^.

**Figure 4 molecules-27-03968-f004:**
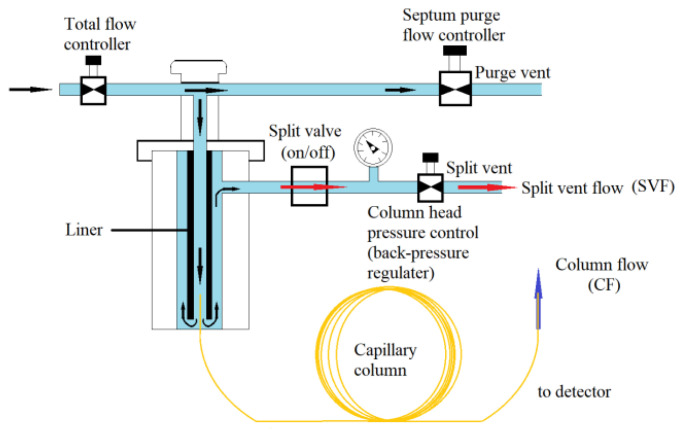
Schematic diagram of split/splitless injection of a GC inlet.

**Figure 5 molecules-27-03968-f005:**
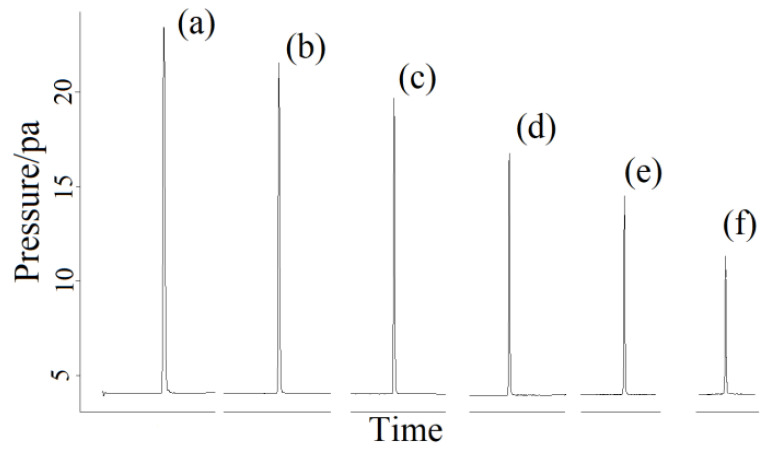
The average chromatographic peak of methane with different split ratios (1:1, **a**; 2:1, **b**; 3:1, **c**; 4:1, **d**; 5:1, **e**; 6:1, **f**) (5 repetitions per sample) under manual split mode.

**Figure 6 molecules-27-03968-f006:**
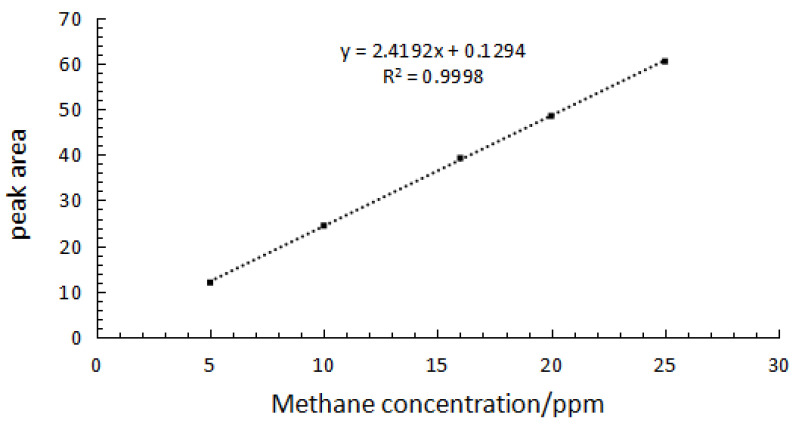
The working curve of methane content determination.

**Figure 7 molecules-27-03968-f007:**
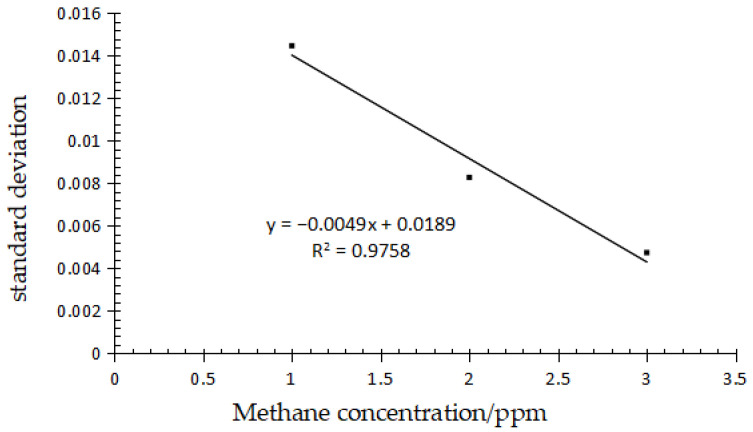
Standard deviation diagram for each concentration.

**Figure 8 molecules-27-03968-f008:**
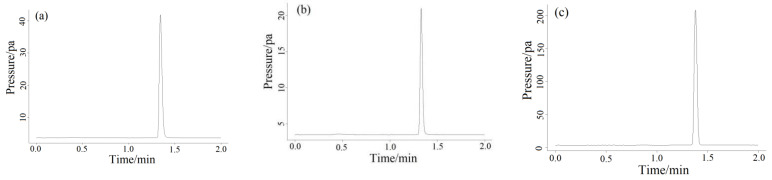
Three samples of methane gas produced by low-methane rice were randomly selected for detection. The detected concentrations of methane in (**a**) (F treatment), (**b**) (O treatment) and (**c**) (AC treatment) were 33.04208, 12.61878 and 172.73617 ppm, respectively.

**Figure 9 molecules-27-03968-f009:**
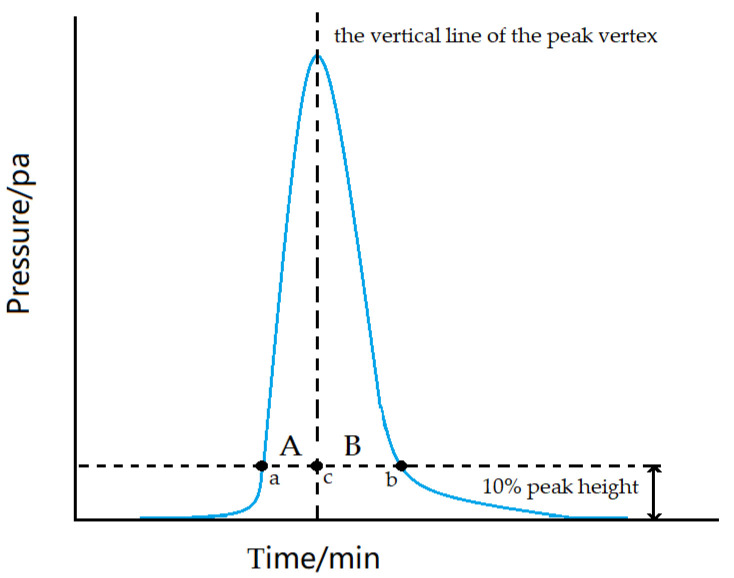
Schematic diagram of symmetry factor. The horizontal and vertical dashed lines are the 10% peak height line and the peak vertex vertical line respectively, intersecting at point c. The horizontal dotted line crosses the blue chromatogram at point a on the left and point b on the right. The distance from a to c is A, and similarly, the distance from c to b is B. The value of A divided by B is the magnitude of the peak symmetry factor.

**Table 1 molecules-27-03968-t001:** RSD and Recycle rate.

Standard Content (ppm)	Measured Content (ppm)	RSD (%)	Recycle Rate (%)
5	4.98369	2.19%	99.67
10	10.16944	1.60%	101.69
16	16.32363	2.71%	102.02
20	20.16350	2.82%	100.82
25	25.32475	1.01%	101.30

**Table 2 molecules-27-03968-t002:** Limit of detection (LOD) and limit of quantification (LOQ).

Gas Concentration (ppm)	Standard Deviation (%)	S_0_	3S_0_ (LOD)	10S_0_ (LOQ)
1	1.44690%	0.0189	0.0567	0.189
2	0.82748%			
3	0.47354			

## Data Availability

Data is contained within the article.

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
