# Peer review of "Low-Cost Detection of Methane Gas in Rice Cultivation by Gas Chromatography-Flame Ionization Detector Based on Manual Injection and Split Pattern"

_molecules, 2022, doi:10.3390/molecules27133968_

Round 1

Reviewer 1 Report

Comments to authors:

The authors submitted the manuscript entitled "Accurate detection of methane gas in rice cultivation by gas chromatography-flame ionization detector based on manual injection and split pattern" to be able to publish at molecules.

Generally, this manuscript needs to improve scientific English writing. The authors may consider to get editing help from someone with full professional proficiency in English. 

The authors suggested an accurate GC method fordetection of methane gas in rice cultivation.In fact, the work is very interesting but the authors need to be presented the experiment and results in a very clear way.

The manuscript still needs to be revised and improved in order to be able for publication consideration. So that I cannot recommend the publication of the manuscript in its actual state.The major revisionis recommended.

Some comments are below:

  • LOD and LOQ should be presented in the abstract.
  • RSD should be presented with one value not a range.
  • Also, the real samples detected should be mentioned in the abstract.
  • The authors should explain how is methane emission from rice fields. And what are the treatments.
  • Authors also need to mention about the types of rice treatment under their studies. (oxantel pamoate treatment (O treatment), fumaric acid treatment (F treatment) and alcohol treatment (AC treatment)).
  • What are the acceptable limits of methane in rice?
  • The authors should mention whether the real samples contained methane within acceptable limit or not?
  • The reference below is much helpful for presenting the work:

SCIENTIFIC REPORTS, (2020) 10:5214. https://doi.org/10.1038/s41598-020-62149-x

Pharmaceuticals 2020, 13, 412;  https://doi.org/10.3390/ph13110412

Reviewer 2 Report

in my opinion: The work is interesting from an analytical point of view. Methane analysis by manual injection is now only found in university laboratories. Currently, automatic analyzers (eg Mettler-Toledo) are predominantly on the market. The work is technically correct. 

In Fig. 8, where the authors present the SD results, it can be seen after conversion that the RSD for a concentration of 4 ppm is 2.7%, for a concentration of 8 ppm it is 2.6% and for a concentration of 16 ppm it is 1.9%. Due to the title of the work, that it is a precise method, I do not agree with these results.

Before I finally write my opinion, I must look at the results obtained to create the calibration curve. I am asking the authors to send to me the original data with methane surface areas with triplicate replicates for each of them. Please send me the results already received. Please don't make new results.

Round 2

Reviewer 1 Report

The authors answer all comments in a reasonable way and modified the manuscript as suggested. So, I recommend the acceptance of the manuscript as it is.

Reviewer 2 Report

Accept in present form